# Multivalency drives interactions of alpha-synuclein fibrils with tau

**Jennifer Ramirez**[1], **Ibrahim G. Saleh**[2], **Evan S. K. Yanagawa**[2], **Marie Shimogawa**[2], **Emily Brackhahn**[2], **E. James Petersson**[1,2]\*, **Elizabeth Rhoades**[1,2]\*

**1** Biochemistry and Molecular Biophysics Graduate Group, Perelman School of Medicine, University of Pennsylvania, Philadelphia, Pennsylvania, United States of America, **2** Department of Chemistry, University of Pennsylvania, Philadelphia, Pennsylvania, United States of America

\* ejpeterss@sas.upenn.edu (EJP); elizabeth.rhoades@sas.upenn.edu (ER)

## Abstract

Age-related neurodegenerative disorders like Alzheimer's disease (AD) and Parkinson's disease (PD) are characterized by deposits of protein aggregates, or amyloid, in various regions of the brain. Historically, aggregation of a single protein was observed to be correlated with these different pathologies: tau in AD and α-synuclein (αS) in PD. However, there is increasing evidence that the pathologies of these two diseases overlap, and the individual proteins may even promote each other's aggregation. Both tau and αS are intrinsically disordered proteins (IDPs), lacking stable secondary and tertiary structure under physiological conditions. In this study we used a combination of biochemical and biophysical techniques to interrogate the interaction of tau with both soluble and fibrillar αS. Fluorescence correlation spectroscopy (FCS) was used to assess the interactions of specific domains of fluorescently labeled tau with full length and C-terminally truncated αS in both monomer and fibrillar forms. We found that full-length tau as well as individual tau domains interact with monomer αS weakly, but this interaction is much more pronounced with αS aggregates. αS aggregates also mildly slow the rate of tau aggregation, although not the final degree of aggregation. Our findings suggest that co-occurrence of tau and αS in disease are more likely to occur through monomer-fiber binding interactions, rather than monomer-monomer or co-aggregation.

**Data Availability Statement:** All raw FCS curves and aggregation data are deposited on figshare https://doi.org/10.6084/m9.figshare.c.7228426.v1.

## Introduction

Alzheimer's disease (AD) and Parkinson's disease (PD) are linked to the accumulation of aggregates of tau and α-synuclein (αS), respectively. Natively, tau is a microtubule-associated protein found predominantly in the axons of neurons that plays a critical role in microtubule stabilization and axonal transport [1], while αS is expressed in the presynaptic termini with a variety of putative functions, including regulation of synaptic vesicles pools [2], neurotransmitter release [3], SNARE complex assembly [4], and vesicle trafficking [5].

Aggregates of αS have been detected in disorders with tau pathology such as AD, and tau filamentation has been observed in synucleinopathies such as PD [6, 7]. In transgenic mouse

**Funding:** This research was supported by the National Institutes of Health (NIH R01 NS103873 to E.J.P., R01 NS120625 to E.R., and RF1 NS125770 to E.J.P. and E.R.). https://www.nih.gov/ Instruments supported by the NIH include a matrix-assisted laser desorption ionization mass spectrometer (S10 OD030460). J.R. was supported by the NIH Chemistry Biology Interface Training Program (T32 GM133398) https://researchtraining.nih.gov/programs/training-grants/T32-a. Sponsors or funders did not play any role in the study design, data collection and analysis, decision to publish, or preparation of the manuscript.

**Competing interests:** The authors have declared that no competing interests exist.

models where tauopathy and synucleinopathy coexist, there is evidence of protein aggregation as well as cognitive and motor deficits [8]. This observation led to the proposal of a synergistic effect between the two pathological proteins on neurodegeneration. *In vivo* models of mutant P301L tau rats exhibit elevated levels of phosphorylated αS, as well as motor dysfunction [9]. Additionally, mice overexpressing K396I tau exhibit L-dopa sensitive Parkinsonism [10]. This overlap in pathology translates to clinical outcomes where these patients have more rapid neurological declines as well as shortened lifespans [11].

Studies using in vitro fibrillization monitored by K114 fluorometry followed by sedimentation analysis have also shown that under some conditions, these two proteins can promote each other's aggregation, further adding to the complexity of these pathologies [12]. However, the molecular basis for their interaction remains unknown.

Tau is comprised of four major domains: the N-terminal projection domain that protrudes from the microtubule surface, the proline rich region (PRR), the microtubule-binding region (MTBR), and the C-terminal domain (Fig 1) [13]. Within the MTBR, there are four weakly conserved repeat sequences, R1-R4. In the adult brain, alternative splicing results in six tau isoforms, based on the absence or presence of inserts (N1, N2) in the N-terminal region and the R2 repeat in the MTBR. Nomenclature for full-length tau is based on the inclusion of these regions. To illustrate, the longest tau isoform contains both N-terminal inserts and all four MTBR repeats and is named 2N4R. For our studies, we used 1N4R, containing the N1 N-terminal insert and all four MTBR repeats and (Fig 1), as this is one of the most abundant isoforms in the adult brain [14].

αS is a 140 amino acid protein consisting of three major domains: the N-terminal domain containing imperfect repeats of the consensus sequence (KTKEGV), the hydrophobic non-amyloid-beta component or (NAC) domain, and the C-terminal domain, enriched in negative charge and proline residues [15, 16] (Fig 1). The first ~100 residues of αS mediate binding to lipid bilayers [17]. In SH-SY5Y cells co-transfected with both tau and αS, an interaction was

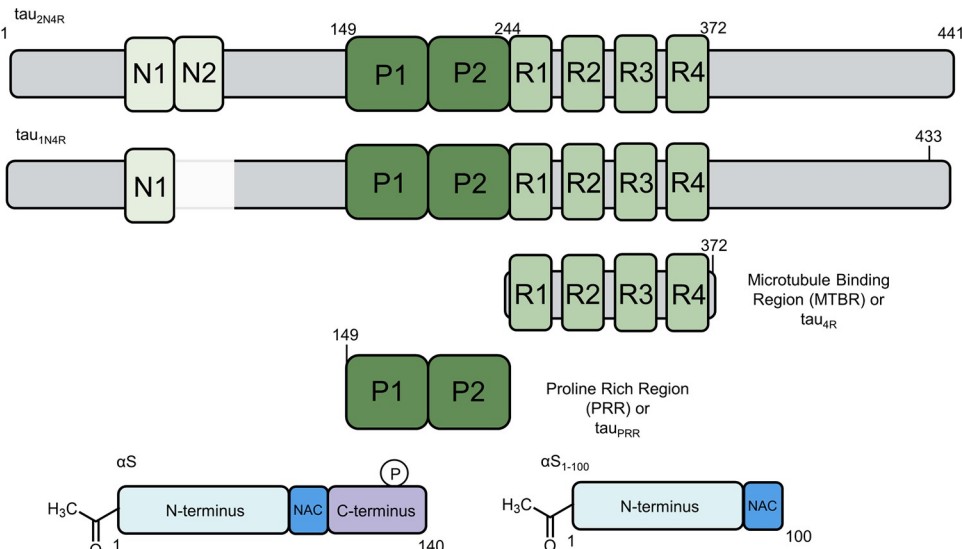

**Fig 1. Schematic of tau and αS constructs.** (upper) Full length tau$_{2N4R}$ and tau$_{1N4R}$ (used in this study) constructs with alternatively spliced N-terminal inserts, N1 and N2, the proline rich region (PRR) and the microtubule binding region (MTBR) with all four repeats. Numbering of tau constructs is based on tau$_{2N4R}$ (1–441). (center) Tau fragments, tau$_{4R}$ and tau$_{PRR}$ used in this study, with Alexa488 fluorophore labeling sites indicated by tick marks. (lower) Full-length αS and αS$_{1-100}$, with the major domains are labeled: The N-terminus, the NAC domain, and the C-terminus. Phosphorylation at serine 129 is indicated as a P on full-length αS.

detected between the two proteins via a Bimolecular Fluorescence Complementation assay [18]. Another study used NMR to map interactions between the C-terminus of αS and tau 1N3R as well as a tau fragment corresponding to 3R alone [19]. Further evidence for the importance of the C-terminus of αS in interactions with tau comes from the observation that successive deletion of C-terminal residues reduced the amount of fibrillar αS bound to monomer tau, while the same treatment of N-terminal residues did not impact binding [20]. This same study mapped binding of αS to tau's central region, including PRR and MTBR [20].

Here, we investigate the interactions of monomer tau with monomer and fibrillar αS. To determine the roles of various domains of both proteins, we use fragments of tau consisting of the PRR ($tau_{PRR}$), the MTBR with R1-R4 ($tau_{4R}$, also referred to as K18 in the literature [21]) and full-length 1N4R, $tau_{1N4R}$. For αS, we use N-terminally acetylated protein, either full-length or truncated at residue 100, $αS_{1-100}$. We also generate N-terminally acetylated αS phosphorylated at serine 129, αS pS129 (Materials and Methods). N-terminally acetylated αS is the physiological form of the protein [22] while nearly all Lewy bodies include αS with phosphorylation at Ser129 [23]. The interaction between tau and αS is measured by fluorescence correlation spectroscopy (FCS), complimented with ensemble aggregation studies. We determine that interactions between tau and monomer αS are very weak. In contrast, tau interacts much more strongly with fibrillar αS; consequently, αS aggregates inhibit tau aggregation, although they do not impact the amount converted into fibrils. As a whole, our results suggest that co-aggregates in disease may arise by interactions between oligomeric or fibrillar species, rather than monomers, or that other cellular factors may be required for co-aggregation.

## Materials and methods

### Tau cloning, purification, and labeling

Tau sequences used were cloned into a lab-made pET-HT vector (a gift from L. Regan). All tau sequences contain an N-terminal His-tag with a tobacco etch virus (TEV) protease cleavage site. For site specific labeling for FCS, the native cysteines at positions 291 and 322 (numbering based on full-length tau protein 2N4R [UniProt:P10636-8]) were mutated to serine, and a non-native cysteine was introduced at position 372 for $tau_{4R}$ and position 433 for $tau_{1N4R}$. The sequence for $tau_{PRR}$ does not contain a cysteine so one was introduced at position 149. Growth and purification of tau constructs were based on methods described previously [24]. A 1 L tau expression for $tau_{1N4R}$ and 0.5 L expressions for $tau_{PRR}$ and $tau_{4R}$ in LB broth (Miller) supplemented with 100 μg/mL ampicillin were induced with 1 mM isopropylthio-β-galactoside (IPTG) at OD ~0.6 for 4 hours at 37°C. The culture was then pelleted at 4600 rpm for 20 minutes at 4°C. The pellet was resuspended in 30 mL of Ni-NTA tau Buffer A (TBA: 50 mM Tris pH 8, 500 mM NaCl, 10 mM imidazole) with 1 mg/mL chicken egg-white lysozyme (Sigma), 1 tablet of EDTA-free protease inhibitor tablet (Roche), and 1 mM phenylmethylsulfonyl fluoride (PMSF). The resuspended pellet was sonicated on ice for 1 minute and 40 seconds, 1 second on/2 seconds off, power set to 50 W. The cellular debris was removed by centrifugation at 20,000 x g for 30 minutes. The supernatant (after centrifugation) was filtered with a 0.22 μM syringe filter and added to 5–7 mL of Ni-NTA resin equilibrated with TBA, followed by incubation with rocking at 4°C for ~1 hour. The resin was then washed with ~30 mL TBA and the protein was eluted with ~15 mL Ni-NTA tau Buffer B (TBB: 50 mM Tris pH 8, 500 mM NaCl, 400 mM imidazole). The eluate was then exchanged back into TBA and concentrated to ~1 mL using Amicon concentrators (Sigma). The His-tag was cleaved by overnight incubation at 4°C with TEV protease (100 μL of 260 μM added to 1 mL tau) and freshly prepared 1 mM dithiothreitol (DTT). Removal of the cleaved His-tag, as well as any uncleaved His-tagged tau, was achieved by incubation of the cleaved sample with TBA equilibrated Ni-NTA resin for 1

hour at 4°C with rocking. The column flow-through containing the cleaved tau protein was exchanged into tau Buffer C (TBC; 25 mM Tris PH 8, 100 mM NaCl) using an Amicon concentrator and concentrated down to 0.5–2 mL. The solution was filtered using a 0.22 μm syringe filter and further purified on an ÄKTA pure FPLC system using a HiLoad Superdex 200 pg size exclusion column.

For generating labeled constructs, freshly purified tau was reduced by incubation with 1 mM DTT for 10 minutes. DTT was removed by exchanging the sample into labeling buffer (20 mM Tris pH 7.4, 50 mM NaCl, 6 M guanidine HCl) using Amicon filters. Alexa Fluor 488 maleimide in DMSO was added in five-fold molar excess to protein and incubated overnight at 4°C with stirring. Labeled protein was exchanged into 20 mM Tris pH 7.4, 50 mM NaCl buffer, and unreacted dye was removed by passing the solution over two coupled HiTrap Desalting columns. Following purification, the samples concentration was determined using a Nano-Drop$^{TM}$ spectrophotometer using the following extinction coefficients: Alexa 488 $\varepsilon$@494 nm = 73000 M$^{-1}$cm$^{-1}$, tau$_{1N4R}$ $\varepsilon$@280 nm = 7450 M$^{-1}$cm$^{-}$1 or tau$_{4R/PRR}$ $\varepsilon$@280 nm 1490 M$^{-1}$cm$^{-1}$ respectively, and calculated by correcting the absorbance signal at 280 nm by Alexa 488 using the following equation: A(tau)$_{280}$ = [A$_{280}$-0.11(A$_{494}$)]/A$_{280}$). The samples were then were aliquoted into microcentrifuge tubes, flash frozen, and stored at -80°C.

## αS cloning and purification

N-terminally acetylated αS [UniProt: P37840] was produced by co-transfection of BL21 cells with both αS and N-terminal acetyltransferase B (NatB; a gift from D. Mulvihill) plasmids, using ampicillin and chloramphenicol to select for colonies containing both plasmids [25].

The parent αS plasmid containing αS fused to a His-tagged GyrA intein from Mycobacterium xenopi (αS-intein) was expressed and purified as previously described for non-acetylated αS [26, 27]. A 0.5 L LB broth was supplemented with 100 μg/mL ampicillin and 34 μg/mL chloramphenicol and induced with 1 mM IPTG at OD ~0.6 overnight at 16°C. The culture was centrifuged at 4600x g for 20 minutes at 4°C. The pellet was resuspended in 30 mL of 40 mM Tris pH 8 supplemented with 1 tablet of EDTA-free protease inhibitor tablet (Roche), and 0.1 mM PMSF. The resuspended pellet was then sonicated on ice for 1 minute and 40 seconds, 1 second on/2 seconds off, power set to 50 W. The cellular debris was removed by centrifugation at 20,000 x g for 30 minutes. The supernatant (after centrifugation) was filtered with a 0.22 μM syringe filter and added to 5–7 mL of Ni-NTA resin equilibrated with αS-intein buffer 1 (ASB1; 50 mM HEPES pH 7.5), followed by incubation with rocking at 4°C for ~1 hour. The resin was then washed with ~15 mL ASB1, followed by a wash with αS-intein buffer 2 (ASB2; 50 mM HEPES pH 7.5, 5 mM imidazole). The protein was eluted with ~12 mL Ni-NTA αS-intein buffer 3 (ASB3: 50 mM HEPES pH 8, 400 mM imidazole). To cleave the His-tagged intein, β-mercaptoethanol (BME; 200 mM final concentration) was added to the eluate and incubated overnight (16–18 hours) at room temperature with rocking. The sample was dialyzed against 20 mM Tris pH 8.0 at 4°C with three changes of the dialysis buffer. Removal of the cleaved His-tagged intein, as well as any un-cleaved αS-intein, was achieved by incubation of the cleaved sample with Ni-NTA resin equilibrated with 20 mM Tris pH 8.0 for 1 hour at 4°C with rocking. The column flow through (containing cleaved αS) was filtered using a 0.22 μm filter and further purified using a 5 mL HiTrap Q HP where it elutes with ~ 300 mM NaCl.

The αS pS129 was generated and purified as described above with the inclusion of a kanamycin-resistant plasmid encoding for polo-like kinase 2 (PLK2) (gift from D.T.S. Pak), a kinase that targets serine 129 [28]. Phosphorylation was confirmed via mass shift (S1 Fig) with MALDI-TOF using a Bruker rapifleX.

C-terminally truncated αS (αS$_{1-100}$) was expressed from a T7-7 plasmid with a stop codon inserted following residue 100 [29]. As with full-length αS, a NatB co-expression plasmid was used to generate N-terminally acetylated αS$_{1-100}$. A culture of 0.5 L LB broth supplemented with 100 μg/mL ampicillin and 34 μg/mL chloramphenicol was induced with 320 μM IPTG at OD ~ 0.6 and grown overnight at 16˚C. The culture was then centrifuged at 4600x g for 20 minutes at 4˚C and the resulting pellet was resuspended in 25 mL lysis buffer (20 mM Tris pH 8.0, 40 mM NaOH, 1 mM EDTA, 0.1% Triton X-100, 1mM PMSF) supplemented with 1 tablet of protease inhibitor cocktail (Roche), followed by the addition of 250 μL 1 M MgCl$_2$, 250 μL 1 M CaCl$_2$, and 40 μL DNase (Roche; 400U total). The sample was incubated at 37˚C for 1 hour with 250 rpm shaking. Following incubation, 500 μL of 0.5 M EDTA was added and cellular debris was removed by centrifugation at 16,900xg for 15 minutes. Ammonium sulfate cuts were used (0.116 g/mL and 0.244 g/mL) to precipitate αS in the second step. The pellet was resolubilized in αS Buffer A (ASBA; 25 mM Tris pH 8.0, 20 mM NaCl, 1mM EDTA) with 1 mM PMSF and dialyzed against ASBA overnight to remove ammonium sulfate. Dialyzed samples were then loaded onto a 5 mL HiTrap Q HP anion exchange column and eluted with αS Buffer B (ASBB; 25 mM Tris pH 8.0, 1 M NaCl, 1mM EDTA), where it elutes with ~ 300 mM NaCl. Fractions containing αS$_{1-100}$ were pooled and concentrated using Amicon concentrators, then loaded onto a HiLoad Superdex 200 pg size exclusion column for further final purification. Following purification, the samples were aliquoted out into microcentrifuge tubes, flash frozen and stored at -80˚C.

## αS aggregation

Fibrils of αS and αS$_{1-100}$ were generated by incubation of 100 μM protein for 7 days with agitation at 1300 rpm and 37˚C in 1x PBS (1 mM KH$_2$PO$_4$, 3 mM Na$_2$HPO$_4$, 155 mM NaCl pH 7.4). We estimate that at least ~90% of the protein is converted to insoluble fibrillar species [30]. Fibrils of full length αS and αS$_{1-100}$ were imaged by transmission electron microscopy (TEM, described below) to ensure subsequent seeds formed from fibrillar samples (S2A Fig). Fibrils were stored for later use by freezing aliquots over dry ice for storage at -80˚C. Prior to use, 40 μL aliquots of resuspended fibrils were added to 360 μL of buffer (1:10 dilution, final concentration αS = 10 μM monomer units) in a microcentrifuge tube placed on ice. The diluted sample was sonicated using a QSonica microtip for 2 minutes, amplitude 50, 1 sec on /1 sec off to generate short fibers, termed seeds. Full-length αS seeds were imaged post sonication (S2B Fig). For the seeds, fibers are not separated from soluble species and thus the 'seeds' are expected to be a mix of monomer, oligomer and fiber (S2C Fig). The amount of soluble protein in the fibril seeds pre- and post-sonication was also quantified (S2D Fig). Based on our gel quantification, (S2C Fig) prior to sonication, we estimate that the majority of the protein for samples of full-length αS and αS$_{1-100}$ is insoluble, and after sonication, approximately 25% of αS is insoluble, while for αS$_{1-100}$, 67% is insoluble. We expect primarily fibrillar species to be present in the pellet, while smaller species, such as monomer and oligomer, are not pelleted during centrifugation. We observed a double band for full-length αS, possibly due to degradation of the protein over the course of its aggregation into fibrils (S2C Fig). For aggregation experiments, the seeds (containing mixtures of monomer, oligomer and fiber) were used immediately; 50 μL of the sonicated material was mixed with 50 μL tau for a final αS seed concentration of 5 μM (monomer units).

## FCS measurements

FCS measurements were conducted on our home-built instrument as described previously [31]. Briefly, the power of a 488 nM DPSS laser was adjusted to ∼ 5 μW [29, 32, 33] as

measured prior to entering the microscope. Fluorescence emission was collected through the objective and separated from laser excitation using a Z488RDC long-pass dichroic and an HQ600/200M bandpass filter and focused onto the apertures of a 50 μm diameter optical fiber directly coupled to an avalanche photodiode. A digital correlator (FLEX03LQ-12, http://correlator.com) was used to generate the autocorrelation curves. All FCS measurements were carried out at ~20 nM fluorescently labeled protein at 20°C in Nunc chambered coverslips (ThermoFisher). The chambers were pre-incubated with poly-lysine conjugated polyethylene glycol (PLL-PEG) to minimize non-specific adsorption of the proteins [34]. All FCS measurements were made in 50 mM $Na_2HPO_4$, 50 mM NaCl pH 7.0 buffer, which was previously used in NMR titration experiments between αS and tau [19]. Samples with αS seeds were measured within 30 minutes of preparation as they appeared unstable at longer time points.

For each measurement of tau construct alone or in the presence of monomer αS, 25 traces of 10 s were obtained. All raw FCS curves are deposited on figshare https://doi.org/10.6084/m9.figshare.c.7228426.v1. The autocorrelation function $G(\tau)$ was calculated as a function of the delay time $\tau$ and then fit using lab-written scripts in MATLAB to a single-species diffusion equation (Eq 1):

$$G(\tau) = \frac{1}{N} \times \frac{1}{1 + \frac{\tau}{\tau_D}} \times \sqrt{\frac{1}{1 + \frac{s^2\tau}{\tau_D}}}$$

where $N$ is the average number of molecules in the focal volume, $s$ is the ratio of radial to axial dimensions of the focal volume (s = 0.2 for our system, as determined by calibration with Alexa 488 fluorophore), and $\tau_D$ is the translational diffusion time. A single component fit was used due to the expected homogenous nature of the monomer sample in presence of a single dilute species, in this case the tau constructs. These curves showed no evidence of a photophysical triplet state nor of unreacted fluorophore affecting the final results (S3 Fig). Infrequently, the presence of large bright species resulting from impurities or non-specific protein assemblies can disproportionally weigh the averaged autocorrelation curves used in the analysis described above (S4 Fig). Moreover, parameters obtained from fitting of these outliers inflates the average $\tau_D$ values used in our analysis. To eliminate these outliers, we evaluated the goodness of fit of the model above as follows: for every set of curves, the summed squared residuals (SSR) from each autocorrelation curve and its fit were calculated; the individual SSR values were averaged and curves with SSR more than three standard deviations from average were discarded due to poorness of fit. This process resulted in discarding <0.3% of the curves for analysis in all conditions. The number of aberrant curves discarded for each condition is listed in S1 Table. To account for day-to-day differences in the alignment of the instrument, the $\tau_D$ values obtained were normalized to the $\tau_D$ values of the tau construct alone. $\tau_{D,norm}$ and standard deviation were calculated for a minimum of three technical replicates per tau construct at its respective concentration with αS monomer. Means were compared for statistical difference (S2 Table) using one-tailed t-tests for comparing each tau or eGFP alone to 150 μM αS (GraphPad Prism).

For each measurement of tau in the presence of αS aggregates, 25 traces of 10 s were obtained. A minimum of three technical replicates per tau construct at its respective concentration with αS aggregates were conducted. All raw FCS curves are deposited on figshare https://doi.org/10.6084/m9.figshare.c.7228426.v1. The autocorrelation function $G(\tau)$ was calculated as a function of the delay time $\tau$ and then fit using lab-written scripts in MATLAB to a two-component species diffusion equation (Eq 2):

$$G(\tau) = \frac{1}{N}\left( A \times \frac{1}{1 + \frac{\tau}{\tau_{D1}}} \times \sqrt{\frac{1}{1 + \frac{s^2\tau}{\tau_{D1}}}} + Q(1-A) \times \frac{1}{1 + \frac{\tau}{\tau_{D2}}} \times \sqrt{\frac{1}{1 + \frac{s^2\tau}{\tau_{D2}}}} \right)$$

where G($\tau$) is the autocorrelation function, N is the number of molecules in the focal volume, $\tau_{D1}$ is the characteristic diffusion time of the tau construct, $\tau_{D2}$ is the characteristic diffusion time of the tau bound to $\alpha$S aggregates, *s* is the ratio of radial to axial dimensions of the focal volume, Q is the 'brightness' of the bound tau species (relative to monomer tau), and A is the fraction of free tau. For fitting of autocorrelation curves in the presence of $\alpha$S seeds, the diffusion time of free tau, $\tau_{D1,}$ was fixed to the value determined for tau alone fit with a single-species diffusion model as described above. All other parameters, except for *s* which was also fixed as described above, were allowed to float. The same SSR criteria described above were used to identify and eliminate outliers. Fewer than 1% of the data was discarded for all conditions as listed in S1 Table. $\tau_{D2}$ represents the average properties of the bound tau species; here, we use it as means of distinguishing between different classes of diffusing species, without assigning a quantitative assessment of species size/molecular weight. Because we imposed no constraints on $\tau_{D2}$ values, values ranged from indistinguishable from $\tau_{D1}$ [35] (i.e. absence or very weak interaction between tau and $\alpha$S seeds) to values >10 ms (i.e. interaction with very large $\alpha$S seeds), which could not be meaningfully assessed by the 10 s data collection timeframe [36]. Thus, while we discuss all regimes of $\tau_{D2}$ values obtained from the fits in the text and note the fraction of data that is ascribed to each category in S3 Table, we primarily analyze data with $\tau_{D2}$ values falling between these extremes, labeling them as $\tau_{D2app}$ since they cannot be attributed to any clearly defined protein state. While we acknowledge that the aggregates are heterogenous, we assigned medians to our $\tau_{D2app}$ values as a semi-quantitative measure since the median is not influenced by extreme values to the same extent as the mean, and is a robust measure of central tendency for asymmetric data distributions.

## Aggregation assays

Aggregation of 1N4R was carried out with 25 μM tau, 500 μM DTT, and 50 μM Thioflavin T (ThT) in 1x PBS buffer, pH 7.5 in a sample volume of 100 μL. Conditions with seeded $\alpha$S (either full-length or $\alpha$S$_{1\text{-}100}$) included 5 μM $\alpha$S seed (monomer concentration). As mentioned above, for the seeds, fibers are not separated from soluble species and thus the 'seeds' are expected to be a mix of monomer, oligomer and fiber (S2 Fig). All components were mixed in a microcentrifuge tube and then transferred to a half-area 96 well plate (Grenier). The aggregation measurements were conducted using the Tecan Spark plate reader, with continuous orbital shaking at a frequency of 510 rpm within the plate reader at 37°C. Aggregation progress was monitored over 120 hours, and ThT fluorescence was measured with bottom mode at an excitation wavelength, $\lambda_{ex}$ = 430 nm (bandwidth 30) and emission wavelength, $\lambda$em $=$ 485 nm (bandwidth 20), 30 flashes with a gain of 40, with time points collected every 15 minutes. All raw aggregation data are deposited on figshare https://doi.org/10.6084/m9.figshare.c.7228426.v1. The data points were analyzed using GraphPad Prism software with a nonlinear regression model [37] described by the equation (Eq 3):

$$y = Min + \frac{Max - Min}{1 + \left(\frac{T_{1/2}}{x}\right)^z}$$

where Min and Max represent the minimum and maximum y values, $T_{1/2}$ is the time at the midpoint of aggregation, and z reflects the curve's steepness. The aggregation time was normalized to the $T_{1/2}$ of the tau$_{1N4R}$ alone, performed concurrently, and minimum and maximum fluorescence values were also normalized. Reported values for aggregation kinetics represent the average and standard deviation calculated from independent triplicate measurements. After the final time point, aggregates were pelleted by centrifugation at maximum speed on a tabletop

centrifuge (13,200 rpm) for 90 min. The aggregated pellet (denoted as P), containing both αS seeds and tau, and the supernatant (denoted as S) containing all soluble material, were analyzed by SDS-PAGE. Samples were supplemented with SDS to 25 mM final concentration, boiled for 20 min, and chilled on ice prior to loading on the gel (4–12% Bis Tris, 200 V, 30 min). We note that under the conditions used (SDS and boiled prior to loading), both tau and αS run predominantly as monomer proteins, irrespective of their initial states, with some higher order assemblies that do not fully dissociate [38]. Gel bands were quantified with ImageJ software. A one-way ANOVA test (GraphPad Prism) was performed to test for any statistical difference in the amount of pellet in the post-aggregation gel samples.

### TEM imaging

Following aggregation, fibril samples were pelleted in the same manner as described above, and the pellets were resuspended in 1X PBS buffer, pH 7.4. The fibrils (5 μL of 7 μM) were added to freshly glow-discharged carbon coated mesh grids and incubated for 2 minutes. Excess buffer was absorbed by wicking with filter paper. To remove excess buffer salts, grids were washed with 5 μL water, again followed by wicking. Staining of the aggregates was achieved by two rounds of incubation with 5 μL uranyl acetate (2% in water), as described above. The grids were dried under vacuum pump for 2 minutes. The post-aggregation samples were imaged with an accelerating voltage of 100kV and a magnification of 26000x on a T12 Tencai microscope.

## Results

The binding of tau to both αS monomers and aggregated seeds was probed using FCS. One advantage of FCS is that it allows for the study of equilibrium protein interactions at low concentrations of fluorescent molecules, such as the nanomolar range used for the tau constructs in our study [32]. Measurements of fluorescently labeled $tau_{1N4R}$, $tau_{4R}$, and $tau_{PRR}$ were made in the absence or presence of unlabeled αS monomer or αS aggregates. We chose to label tau rather than αS since FCS measurements with labeled αS aggregates would have been insensitive to interactions with tau.

### Tau interacts weakly with monomer αS

To measure the interaction between tau and αS, the diffusion time, $\tau_D$, of fluorescently labeled tau was determined with increasing concentrations of monomer αS. A change in the apparent size of the fluorescently labeled species due to tau interacting with αS results in an increase in the measured diffusion time. While, in principle, a population of free tau and αS-bound tau is present in these measurements, it is likely that the difference in diffusion time between the free and bound tau species is not sufficiently large to be reliably distinguished [35]. Therefore, we chose to fit these data with a single diffusing species model (Eq 1), which serves at least as a qualitative measure of the interaction. All three tau constructs interact with monomer αS, as seen by the increase in their diffusion times in its presence (Fig 2A–2C). The increase in $\tau_D$ is very weakly concentration dependent (S5 Fig) and high αS:tau ratios are required to observe an interaction (S4 Table). As a whole, these data do not support a 1:1 binding mode for any of the tau constructs with monomer αS, but rather weak, electrostatically driven interactions that often are found with disordered proteins; as a consequence, our assessment of the increase in $\tau_D$ is more qualitative. We report the percent difference relative to $\tau_D$ in the absence of αS (S4 Table); at the highest concentrations of αS measured, 150 μM, increases of 7, 10, and 27 (upper population) or 7 (lower population) % were measured for $tau_{1N4R}$, $tau_{4R}$, and $tau_{PRR}$ respectively (S4 Table). Applying a one-tailed t-test showed the means were statistically different for

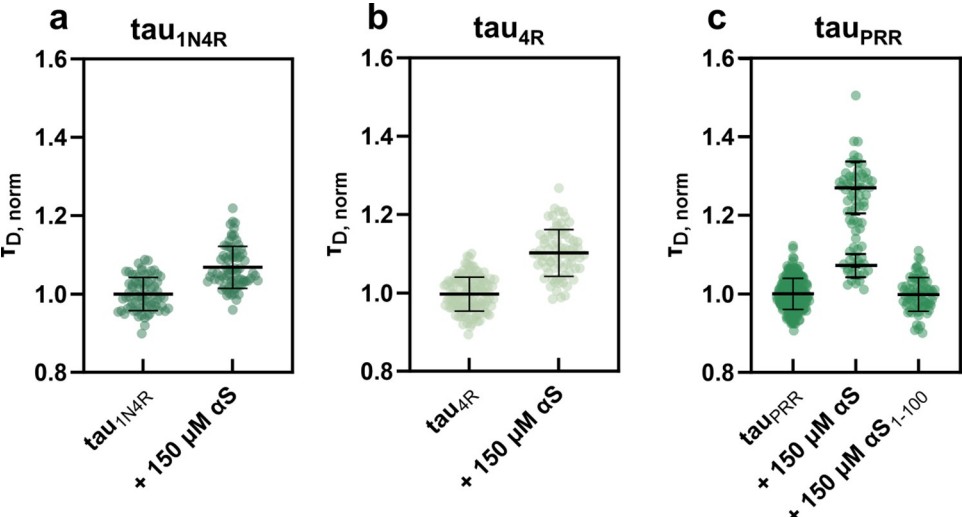

**Fig 2.** Distribution of normalized diffusion times, $\tau_D$, for tau$_{1N4R}$ (a) tau$_{4R}$ (b) and tau$_{PRR}$ (c) in the absence and presence of 150 μM αS. For tau$_{PRR}$, measurements were also carried out in the presence of 150 μM αS$_{1-100}$, with tau concentrations ~20 nM (as approximated by $N$ in Eq 1) to give a 1:7500 molar ratio of tau:αS. Mean $\tau_D$ and SD were calculated for a minimum of three measurements with αS. The values obtained from normalized $\tau_D$ are found in S4 Table, as well as the percent change in the presence of αS relative to tau alone. Significance tests by t-test are recorded in S2 Table.

all three constructs in the presence of 150 μM αS (S2 Table). We used enhanced green fluorescent protein (eGFP), in the absence and presence of 150 μM αS, as a control; since eGFP is not expressed in mammalian cells, it does not represent a possible physiological binding partner. As with the tau constructs, we observed a small increase in $\tau_D$ for eGFP in the presence of high concentrations of αS (S6 Fig). The relative increase in $\tau_D$ was ~4% (S6 Fig), smaller than the values observed for any of the tau constructs and αS. Moreover, while the t-test for eGFP showed a statistical difference in the presence of αS, its t-value was also smaller than those calculated for tau (S2 Table). As a whole, these data suggest that the interactions between the various tau constructs and monomer αS are only slightly stronger than the non-specific ones between eGFP and αS.

Interestingly, tau$_{PRR}$, a domain which had not previously been identified as independently interacting with αS, shows evidence of two populations of tau-αS complexes at the highest αS concentration measured (Fig 2C and S5 Fig), which likely reflects binding of multiple αS per tau$_{PRR}$. Because of its relatively large increase, we used tau$_{PRR}$ as a point of comparison for other αS variants. We found no increase in $\tau_D$ for tau$_{PRR}$ in the presence of 150 μM αS$_{1-100}$ (Fig 2). This observation of no interaction between tau$_{PRR}$ and αS$_{1-100}$ is consistent with prior studies of other tau domains, underscoring the importance of the negatively charged C-terminus of αS in mediating even weak interactions with positively charged tau$_{PRR}$ (Fig 2C).

### αS aggregates enhance interactions with tau

Aggregation of tau can be accelerated through 'seeding', i.e. adding tau, or even other protein, aggregates to a solution of tau monomer [39]. This prompted us to investigate the interaction between tau monomers and αS aggregates. Fibrillar fragments, or seeds, of αS or αS$_{1-100}$ were generated as described in the Materials & Methods and added to labeled tau monomer in increasing concentrations. The resulting autocorrelation curves were fit with Eq 2. As described in the Materials and Methods, $\tau_{D2app}$ values were assigned to one of three

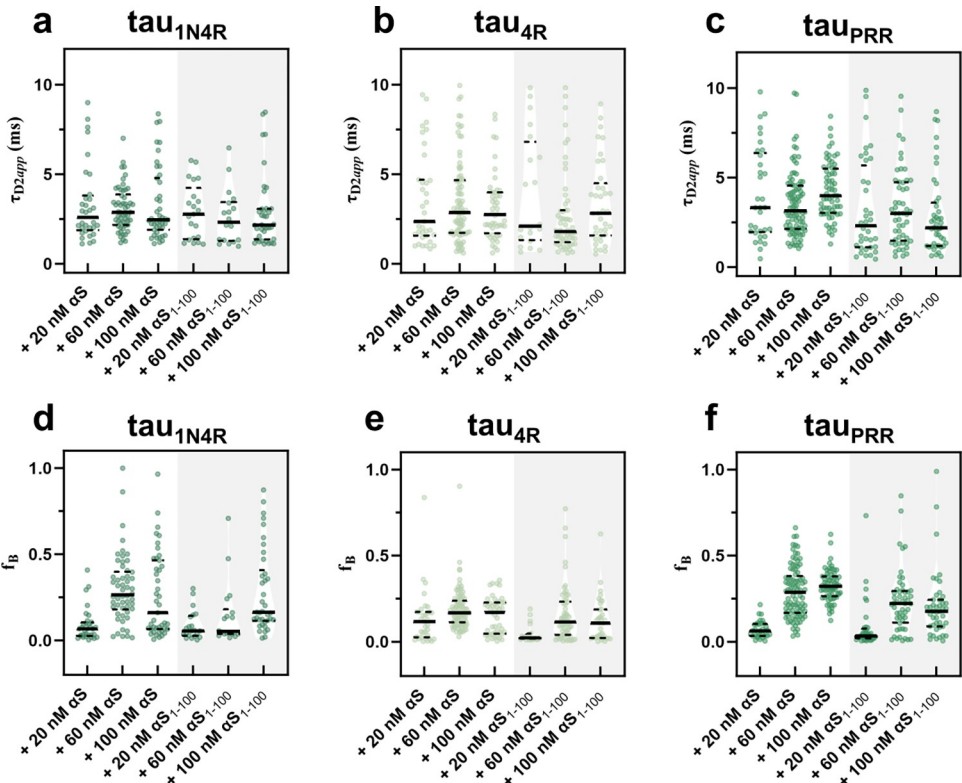

**Fig 3. Individual type III $\tau_{D2app}$ and $f_B$ values.** Data is shown for a) tau$_{1N4R}$, b) tau$_{4R}$, c) tau$_{PRR}$ upon the addition of unlabeled full-length or truncated (shaded) αS seeds. Individual $f_B$ values plotted for d) tau$_{1N4R}$, e) tau$_{4R}$, f) tau$_{PRR}$. These values are summarized in Table 1. Tau concentration ~20 nM give an approximate 1:1 molar ratio of tau:αS (monomer units) is indicated on x-axis labels. The medians (solid black line) and quartiles (dashed black lines, 25 and 75%) are depicted on the plots and values are listed in Table 1. These data are compiled from a minimum of three measurements for each condition.

classifications: (type I) not distinguished from a single diffusing component model ($\tau_{D2}<1.6*\tau_{D1}$) [35]; (type II) too large to be meaningfully assessed ($\tau_{D2}>10$ ms) [36]; or (type III) within the range to be interpreted in terms of the number tau constructs bound and approximate size of the diffusing αS species (10 ms$>\tau_{D2}>1.6*\tau_{D1}$). The type II and type III curves collectively describe interactions between tau and αS oligomers/fibers. The relative fractions of curves assigned to each of these categories for all measurement conditions is summarized in S3 Table.

The distributions of type III $\tau_{D2app}$ values are plotted in Fig 3; in contrast to the $\tau_D$ data for tau interaction with monomer αS, which was well-described by Gaussian distributions, the $\tau_{D2}$ distributions are asymmetric, with long tails of higher $\tau_{D2}$ values (S7 Fig). Distributions of $f_B$, the tau fraction associated with $\tau_{D2}$ ($f_B = 1$-A from Eq 2), are similarly asymmetric (Fig 3). Thus, we used the median values of the fit parameters as a point of comparison between the different tau constructs and their interactions with αS seeds, summarized in Table 1. There is not a significant amount of variation in the median $\tau_{D2app}$ values between tau constructs for different seed concentrations (Table 1). The median values average ~3 ms, suggesting that they primarily represent interactions between tau and smaller aggregates/oligomers formed upon sonication of the fibers. The fraction of signal arising from these larger species increases with increasing αS seed concentration, with higher $f_B$ values at 60 and 100 nM concentrations, compared to 20 nM (Table 1). Both the average value of the median $\tau_{D2app}$ and the median $f_B$ are

**Table 1. Median type III $\tau_{D2app}$, $f_B$ (with associated quartiles, 25% & 75%), and Q values for tau constructs in the absence or presence of different concentrations of αS seeds.**

| Tau construct | $\tau_{D2app}$ | $f_B$ | Q |
|---|---|---|---|
| **tau$_{1N4R}$** | - | - | - |
| + 20 nM αS | 2.6 (1.9, 3.8) | 0.067 (0.026, 0.11) | 1.5 |
| + 60 nM αS | 2.9 (2.2, 3.9) | 0.26 (0.18, 0.40) | 2.7 |
| + 100 nM αS | 2.5 (1.9, 4.8) | 0.161 (0.066, 0.46) | 2.7 |
| + 20 nM αS$_{1-100}$ | 2.8 (1.4, 4.2) | 0.055 (0.027, 0.14) | 2.3 |
| + 60 nM αS$_{1-100}$ | 2.3 (1.3, 3.5) | 0.053 (0.038, 0.18) | 2.0 |
| + 100 nM αS$_{1-100}$ | 2.2 (1.4, 3.1) | 0.16 (0.11, 0.41) | 2.7 |
| **tau$_{4R}$** | - | - | - |
| + 20 nM αS | 2.4 (1.6, 4.7) | 0.12 (0.026, 0.17) | 1.0 |
| + 60 nM αS | 2.9 (1.7, 4.7) | 0.17 (0.11, 0.24) | 1.0 |
| + 100 nM αS | 2.8 (1.7, 4.0) | 0.17 (0.047, 0.23) | 1.0 |
| + 20 nM αS$_{1-100}$ | 2.1 (1.3, 6.8) | 0.023 (0.014, 0.048) | 1.0 |
| + 60 nM αS$_{1-100}$ | 1.8 (1.2, 3.0) | 0.11 (0.040, 0.23) | 1.0 |
| + 100 nM αS$_{1-100}$ | 2.8 (1.6, 4.5) | 0.11 (0.022, 0.19) | 1.0 |
| **tau$_{PRR}$** | - | - | - |
| + 20 nM αS | 3.3 (2.0, 6.4) | 0.062 (0.034, 0.10) | 1.0 |
| + 60 nM αS | 3.2 (2.1, 4.6) | 0.29 (0.17, 0.38) | 2.4 |
| + 100 nM αS | 4.0 (3.0, 5.5) | 0.32 (0.26, 0.38) | 1.6 |
| + 20 nM αS$_{1-100}$ | 2.3 (1.1, 5.7) | 0.033 (0.019, 0.077) | 1.4 |
| + 60 nM αS$_{1-100}$ | 3.0 (1.5, 4.8) | 0.22 (0.11, 0.30) | 1.1 |
| + 100 nM αS$_{1-100}$ | 2.2 (1.2, 3.6) | 0.18 (0.089, 0.24) | 1.0 |

lower for seeds made with αS$_{1-100}$ relative to full-length αS (Fig 3 and Table 1). Strikingly, the brightness, Q, of the bound tau species is higher for both tau$_{PRR}$ (Q$_{avg}$~1.7) and tau$_{1N4R}$ (Q$_{avg}$~2.3) across seed concentrations (Table 1). This suggests that multiple tau constructs are associating with αS seeds. On average, Q is lower for tau$_{PRR}$ (Q$_{avg}$~1.2) for truncated αS seeds compared to full-length seeds, while remaining unchanged for tau$_{1N4R}$ (Q$_{avg}$~2.3). In contrast, Q~1 for tau$_{4R}$ for both full-length and truncated seeds. Interactions between eGFP and full-length αS fibers were also measured; the median $\tau_{D2app}$ value was comparable to that of the tau constructs, but the $f_B$ value was much lower ($f_B$ = 0.02, S5 Table) than for any of the tau constructs under any conditions, further evidence that interactions between eGFP and αS are weak and non-specific.

For the first two $\tau_{D2app}$ categories, a more qualitative assessment is appropriate. For all three tau constructs, the fraction of curves showing little evidence of binding (type I) to full-length αS seeds was highest and comparable in magnitude at the lowest αS seed (20 nM) concentration (S3 Table). The number of type I curves is higher for seeds made with αS$_{1-100}$ than for full-length αS for all three tau constructs as well, indicating reduced interactions with αS seeds lacking the negatively charged tails. The fraction of data classified as type II or type III (reflecting interactions with oligomer/fibrillar seeds) shows a weak concentration dependence for all tau constructs, with a greater fraction assigned to these categories at 60 and 100 nM αS, for both full-length αS and αS$_{1-100}$ seeds (S3 Table). To test the hypothesis that electrostatic contributions drive the interactions between tau and αS seeds, FCS measurements of tau$_{PRR}$ and 60 nM αS seeds were made in both low (5 mM NaCl) and high salt (500 mM NaCl) buffers. In the low and standard salt conditions, the majority of curves (99% and 93%, respectively) were fit by a standard two-component diffusion model (S3 Table). The fraction of curves assigned as type I increased from 2% in our standard buffer (50 mM NaCl) to 45% in the

higher salt buffer (S3 Table), reflecting a significant decrease in bound tau$_{PRR}$. Moreover, for the type III curves, the median $f_B$ decreases in a salt dependent manner from 0.66 to 0.22 to 0.02 with increasing salt. Notably, the median Q value is highest in the lowest salt buffer, suggesting that the low salt results in an increase in the average number of tau molecules bound per αS oligomer (S5 Table).

### αS seeds slow tau aggregation

Given the observed interaction of tau with αS seeds at nM concentrations, we sought to determine how these differences in binding affect aggregation. Aggregation was tracked using thioflavin T (ThT) fluorescence (Materials and Methods) over the course of 120 hours. Samples were centrifuged following aggregation and both pellet and supernatant were visualized by SDS-PAGE. We then compared the quantity of aggregated material in the absence and presence of seeds of αS or αS$_{1-100}$ following 120 hours of incubation (Fig 4). With seeded conditions, we observed a significant increase in the $T_{1/2}$ with both full-length and truncated αS$_{1-100}$ seeds relative to tau$_{1N4R}$, but no significant difference between the two seeded samples (Fig 4B). Additionally, the two seeded samples had lower levels of ThT fluorescence than tau alone. It is difficult to know how to interpret the differences in ThT signals, since a technical limitation of using ThT is its limited sensitivity to early-stage assemblies of tau. It has been shown that although tau protofibrils do bind ThT, the interaction is often weaker than observed for mature fibrils [40]. Post-aggregation samples run on SDS-PAGE gels to determine the relative amount of protein in the pellet show no significant difference under the conditions we tested (Fig 4D). Under the reducing conditions of the gel, tau$_{1N4R}$ runs predominantly as a monomer, but there are also reducing agent resistant, higher order species

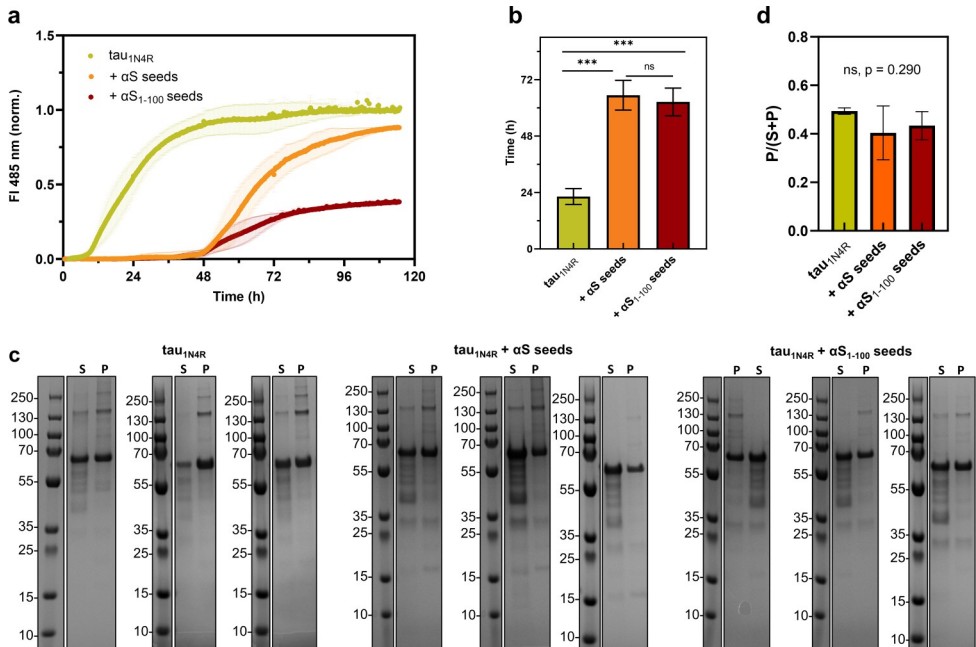

**Fig 4. αS seeds increase aggregation lag time but do not impact the amount of protein pelleted post-aggregation.**
a) ThT aggregation plots of 25 μM tau$_{1N4R}$ alone versus the addition of 5 μM αS and αS$_{1-100}$ seeds (5:1 molar ratio of tau:αS). b) The $T_{1/2}$ times are calculated from aggregation plots. c) SDS-PAGE gels of samples post-aggregation with S indicating the supernatant and P the pellet after centrifugation. d) The amount for each sample was quantified in triplicate using the relative amount of pellet vs. the total amount of pellet and supernatant (ImageJ).

in the supernatant and pellet lanes (Fig 4C), particularly in the pellet lanes. There is also some tau degradation evident by the lower molecular weight bands, likely due to the long aggregation period at warmer temperatures.

The aggregated pellets were also visualized by TEM. All samples showed fibrillar structures (S8 Fig), although morphological differences are observed, further corroborating the overall impact of the αS seeds on tau aggregation under the conditions of our assay.

## Discussion

Since tau and αS were first linked to neurodegenerative disorders, both proteins have been heavily studied separately for their role in diseases by a wide variety of experimental methods, but not as commonly for their roles when both are present. In this study, we have examined the interaction of tau and αS by complementing traditional aggregation experiments with FCS. Our results have determined that 1) full-length tau, as well as individual tau domains, interact with monomer αS weakly; 2) this interaction is much more pronounced for αS aggregates; 3) under the conditions of our assay, this interaction extends the lag time of tau aggregation but does not significantly alter the extent of aggregation. We demonstrate that the PRR-MTBR of tau binds weakly to monomer and oligomer/fibrillar αS (Fig 2); for monomer αS, the C-terminus of αS is required, while for αS seeds, the C-terminus is not essential but enhances the interaction (Fig 2), consistent with prior work emphasizing these regions of both proteins [19, 20]. Our study goes on to show that the PRR and MTBR domains of tau are individually capable of binding to αS. Given the uneven distribution of charged amino acids in both tau and αS (S9 Fig), it is worth considering their impact on the interaction. Both the PRR and MTBR have a comparably high fraction of positively charged residues (~16–17%), and the fraction of negatively charged C-terminus of αS is ~35%. However, in the PRR, there are relatively fewer negatively charged residues, resulting in it having a more positive net charge per residue (NCPR), ~0.143 for PRR to MTBR's ~0.075 (S9 Fig) [41]. This may also explain why the PRR shows the largest shifts in diffusion times in interactions with monomer αS, as well as the largest $f_B$ (100% for 60 and 100 nM seeds) in binding to full-length αS seeds. (Fig 2). We were somewhat surprised to observe that pS129 in αS did not enhance interactions with tau (S10 Fig), as we had hypothesized that the increase in negative charge conferred by the phosphate group would increase binding. However, given the high overall abundance of negative charge in the C-terminus, phosphorylation only moderately increases the NCPR from ~-0.33 to ~-0.35, likely blunting the impact of this post-translational modification on electrostatically driven tau interactions.

Overall, however, electrostatics are clearly important for the interaction between tau and αS seeds, as increasing the buffer salt concentration significantly increased the proportion of FCS curves showing no significant interaction between $tau_{PRR}$ and αS seeds, whereas lowering salt increased the average number of $tau_{PRR}$ bound per seed (S5 Table). Using eGFP as a non-physiological control, we found weak interactions with both monomer and seed αS (S3 and S5 Tables). The net surface charge of eGFP is -7 [42], while the N-terminus of αS is net positively charged. Thus, interactions between these non-physiological partners are also likely electrostatic in nature and represent a threshold for non-specific interactions between tau and αS. That eGFP interacts with αS is not terribly surprising, as the unstructured and highly charged character of αS make it a 'sticky' [43–45] protein. The comparison between tau constructs and eGFP at 60 nM αS seed concentrations shows significantly higher median $f_B$ values for all tau constructs, underscoring greater specificity of these interactions relative to eGFP (Table 1, S5 Table).

Although full-length 1N4R tau includes both the PRR and MTBR, it does not show an additive enhancement in its interactions with monomer αS (Figs 2 and 3). This may be due to the

presence of the net negatively charged N-terminus (NCPR~ -0.143 for the 1N isoform; S9 Fig). In solution, the N-terminus of tau makes long-range electrostatic interactions with the PRR and MTBR [32]. The N-terminus thus may have repulsive interactions with the negatively charged C-terminus of αS, effectively reducing binding of PRR and MTBR to αS in the context of full-length tau. Indeed, tau constructs lacking the N-terminus show enhanced binding to negatively charged tubulin, evidence that N-terminal domain can regulate tau interactions [33].

For all tau constructs, the interaction between tau and αS was significantly enhanced for αS aggregates relative to monomer, with increases in diffusion time seen at significantly lower concentrations (~1000x less than required for monomer αS) of aggregates (Figs 2 and 3). All cryo-electron microscopy and NMR structures of αS fibers to date show the C-terminus of αS on the outside of the fiber, where it remains accessible for binding interactions [46, 47]. As such, the αS fibers display a high density of negatively charged C-termini, providing a multivalent binding site for positively charged biomolecules. Alignment of charged residues along the surface of the fibril core also presents another set of multivalent interaction sites [15, 48], independent of the charged C-terminus. In short, the extensive surface area of the αS fiber may offer numerous interaction sites for tau, thereby enhancing the potential for multivalent interactions. Consistent with this model is our observation that the αS seed bound fraction of $tau_{1N4R}$ was 'brighter' than either $tau_{PRR}$ or $tau_{4R}$ (Q values in Table 1) across all concentrations of seeds and, notably, including seeds made from $αS_{1-100}$. This implies that on average a larger number of $tau_{1N4R}$ bind to each αS seed, suggesting that the presence of both domains enhances the interaction in the context of αS aggregates.

The transient interactions between tau monomers and αS seeds observed by FCS provide a rationale for the increase in tau aggregation lag time in the presence of these seeds. Namely, FCS measurements in solutions with high concentrations of $tau_{1N4R}$ (i.e. comparable in molar ratio to those used for $tau_{1N4R}$ aggregation) showed that even in the presence of five-fold excess tau, the majority of FCS curves (53%, S6 Table) showed evidence of some interaction with oligomer or fibrillar αS. These interactions with αS seeds may act to buffer the monomer tau concentration, rather than provide a surface to template and accelerate tau aggregation. Several other studies have also observed that αS seeds impede the tau aggregation pathway [37], identifying residues 32–57 of αS as crucial components of the fibrillization core for both strains of αS fibrils, influencing their seeding properties [38]. This observation could potentially elucidate why we detect a comparable increase in the $T_{1/2}$ of full-length versus C-terminally truncated αS seeds, as both encompass this region. We do also find interactions between tau and monomer αS, although we did not directly assay whether αS and tau synergistically promote each other's aggregation. This has been reported previously for in vitro aggregation [19], neuronal transduction with αS mouse preformed fibrils leading to tau inclusions that also tested positive for phosphorylated αS [11], and in vivo studies that showed interaction and induction of pathological phosphorylation states for both proteins [18]. However, cellular factors not present in our in vitro studies may have played a role in these studies.

## Conclusions

Our results show that all tau constructs tested here interact weakly with monomer αS, with an enhanced interaction with αS seeds, particularly for $tau_{PRR}$. While the MTBR is often the focus of tau studies, prior work from our lab has drawn attention to the PRR, showing that it has a critical role in binding to tubulin and tubulin polymerization [33], as well as in binding to the aggregation inducer, polyphosphate [49]. Our results also suggest that the mechanism by which αS affects tau pathology is not through directly seeding tau aggregation. For example,

binding of tau to fibrillar αS may sequester tau away from microtubules, increasing the likelihood of it undergoing post-translational modifications or binding to other molecules that lead to aggregation, rather than directly templating tau aggregation. Clearly, more investigation of tau and αS interactions are warranted and our own studies incorporating additional post-translational modifications and cell-based experiments are already underway.

## Supporting information

**S1 Fig. Absorption and MALDI-TOF spectra of αS pS129.** Absorbance spectra a) of acetylated αS and acetylated pS129 αS used to calculated concentrations of both proteins prior to MALDI-TOF ($\varepsilon$@280 nm = 5960 $M^{-1}cm^{-1}$). Both proteins were mixed in equimolar amounts and spotted for b) MADLI-TOF analysis. The relatively equal spectral peak heights of the MALDI-TOF spectra and the expected mass shift indicate that most, if not all, the αS is phosphorylated after the co-expression of NatB and PLK2 plasmids (see Materials and Methods for details).
(PDF)

**S2 Fig. Characterization of αS seeds.** a) Fibrils of full length αS and $αS_{1-100}$ was imaged by transmission electron microscopy (TEM) as described in the Material and Methods to ensure subsequent seeds formed from fibrillar samples. b) Fibrils of full length αS imaged by TEM post sonication as described in Material and Methods. c) SDS-PAGE gel showing the amount of protein present in the supernatant (S) and pellet (P) of full-length αS versus $αS_{1-100}$ fibrils and 'seed' full-length αS versus $αS_{1-100}$ post-sonication as described in Materials and Methods. d) Quantification by ImageJ of average amount of protein based on both gels in the pellet versus the pellet and the supernatant summed shows most of the protein for full-length is in the pellet while for $αS_{1-100}$, 18% remains as soluble species. For the seeds, 75% of full-length αS remains as soluble species while 33% remains for $αS_{1-100}$.
(PDF)

**S3 Fig. Comparison of FCS fitting models.** a) For $tau_{PRR}$, comparison of three fit equations: (1) one diffusing component (protein only); (2) one diffusing component with exponential decay (triplet); (3) two diffusing components (protein and free dye). For all fits, only s was fixed, and all other parameters were allowed to float. b) The equation using the triplet state with the autocorrelation function $G(\tau)$ was calculated as a function of the delay time $\tau$ where $G(\tau)$ is the autocorrelation function, N is the number of molecules in the focal volume, A is the fraction of the exponential contribution, $\tau_T$ is the triplet lifetime, $\tau_D$ is the translational diffusion time, s is the ratio of radial to axial dimensions of the focal volume. The other fit equations are found in the main manuscript. Fit parameters for each model are reported. c) Considering the increased number of free parameters in both the triplet and two component diffusion fits, the fit quality was not improved over the one component fit, as displayed by an F-test showing insufficient evidence to support a difference in the variances of the residuals.
(PDF)

**S4 Fig. Representative individual autocorrelation curves with monomer and seed αS.** Aberrant curves disproportionally weigh the averaged autocorrelation at higher αS concentrations. Representative individual autocorrelation curves using labeled $tau_{4R}$ in the absence (a) or presence of αS monomer (b), full-length seeds (c) or truncated seeds (d). The correlation curves are gray and the fit to the appropriate diffusion equation as described in the Materials and Methods are in red. The plots in the left-hand column display all the collected autocorrelation curves; the plots in the right-hand column display the remaining autocorrelation curves following the SSR analysis described in the Materials and Methods. Very few of the curves (as

quantified in tables below for each construct) fall outside the main distribution, so that discarding aberrant curves with extremely high $\tau_D$ values does not change interpretation of the data.
(PDF)

**S5 Fig. $\tau_D$ for monomer tau$_{4R}$ and tau$_{PRR}$ at increasing concentrations.** Tau binds to $\alpha$S monomer in a weakly concentration dependent manner. Unlabeled $\alpha$S was added to fluorescently labeled tau as described in the Materials & Methods. Normalized $\tau_D$ for a) tau$_{4R}$ and b) tau$_{PRR}$.
(PDF)

**S6 Fig. Interaction of eGFP with monomer $\alpha$S.** eGFP interacts weakly with $\alpha$S monomer. Plots shown $\tau_{D, norm}$ for a) eGFP and b) eGFP with 150 μM $\alpha$S. Mean $\tau_{D, norm}$ and SD calculated for a minimum of three measurements with $\alpha$S. % diff = [$\tau_D$(+$\alpha$S)-$\tau_D$(-$\alpha$S)]/$\tau_D$(-$\alpha$S).
(PDF)

**S7 Fig. Representative distributions of $\tau_D$ values.** Representative distributions of $\tau_D$ values for a) tau$_{PRR}$ and tau$_{PRR}$ plus b) $\alpha$S monomer or c) seed ($\tau_{D2app}$). Both a) and b) show symmetric distributions, while c) is asymmetric with a tail of higher $\tau_{D2app}$ values.
(PDF)

**S8 Fig. Tau aggregates are fibrillar.** Post-aggregation samples for tau$_{1N4R}$ (first panel), followed by samples with $\alpha$S and $\alpha$S$_{1-100}$ seeds, middle and right panel respectively, were imaged on a T12 Tencai microscope.
(PDF)

**S9 Fig. Charge plots of $\alpha$S and tau generated using CIDER.** Charge plots generated using CIDER outline the distribution of positive versus negative charges in the PRR of tau compared to the repeats, as well as the distribution in the domains of $\alpha$S. The numbering for tau is for the longest tau isoform, 2N4R; 1N4R lacks one of the N-terminal inserts (indicated in pink on the plot). CIDER analysis for tau and $\alpha$S constructs are listed in the table, NCPR (net charge per residue), κ (kappa) charge patterning parameter, f+ (fraction of positive residues), and f- (fraction of negative residues).
(PDF)

**S10 Fig. $\tau_D$ with monomer $\alpha$S pS129.** Phosphorylation of monomer $\alpha$S at pS129 does not enhance interactions with tau$_{PRR}$. Unlabeled $\alpha$S was added at concentrations indicated on plot. Both $\alpha$S and $\alpha$S$_{pS129}$ show mild, concentration-dependent increases in binding, reflecting weak interactions. Mean $\tau_{D, norm}$ and SD calculated for a minimum of three measurements with increasing concentration of $\alpha$S. % diff = [$\tau_D$(+$\alpha$S)-$\tau_D$(-$\alpha$S)]/$\tau_D$(-$\alpha$S).
(PDF)

**S1 Table. Analysis of autocorrelation curves discarded through poor fit.**
Table summarizing autocorrelation data. Analysis of initial data sets (column 1) by the SSR as described in the Materials and Methods resulted discarding a few autocorrelation curves (column 2) for each set of measurements. The normalized SSR (column 3) and % curves analyzed (column 4) are also included.
(PDF)

**S2 Table. Results of applying one-tailed t-tests.** Results of one-tailed t-tests (GraphPad Prism) for tau binding to monomer $\alpha$S, ns = not significant, t = t value and df degrees of freedom with a Geisser-Greenhouse correction.
(PDF)

**S3 Table. Classification of seed diffusion times.** Classification of $\tau_{D2app}$ for tau binding to $\alpha$S seeds as described in the main manuscript. Type I: $\tau_{D2} < 1.6^*\tau_{D1}$; Type II: $\tau_{D2} > 10$ ms; Type III: $1.6^* \tau_{D1} < \tau_{D2} < 10$ ms.
(PDF)

**S4 Table. Quantification of binding at 150 μM αS monomer.** Quantification of binding of $tau_{1N4R}$, $tau_{4R}$ and $tau_{PRR}$ with increasing concentrations of $\alpha$S monomer. Mean $\tau_{D, norm}$ and SD calculated for a minimum of three measurements for each concentration $\alpha$S. % diff is calculated as $[\tau_D(+\alpha S)-\tau_D(-\alpha S)]/\tau_D(-\alpha S)$.
(PDF)

**S5 Table. τD2app, Q and fB for GFP and PRR salt concentrations.** Summary median type III τD2app, fB (with associated quartiles, 25% & 75%), and Q values for eGFP and for tauPRR in low salt (5 mM NaCl), standard salt (50 mM NaCl), and high salt (500 mM NaCl) conditions. Minimum of three replicates for each.
(PDF)

**S6 Table. Classification of diffusion times with seeded αS/tau aggregation FCS conditions.** Classification of $\tau_{D2app}$ for $tau_{1N4R}$ binding to $\alpha$S in solutions with high concentrations of $tau_{1N4R}$ (i.e. comparable to those used for $tau_{1N4R}$ aggregation). Type I: $\tau_{D2} < 1.6^*\tau_{D1}$; Type II: $\tau_{D2} > 10$ ms; Type III: $1.6^* \tau_{D1} < \tau_{D2} < 10$ ms.
(PDF)

**S1 Raw image.**
(PDF)

# Acknowledgments

The authors thank Ryan Kubanoff for training on the MALDI-TOF Bruker rapifleX. We acknowledge the use of instruments at the Electron Microscopy Resource Lab.

# Author Contributions

**Conceptualization:** Jennifer Ramirez, E. James Petersson, Elizabeth Rhoades.

**Formal analysis:** Jennifer Ramirez, Ibrahim G. Saleh, E. James Petersson, Elizabeth Rhoades.

**Funding acquisition:** E. James Petersson, Elizabeth Rhoades.

**Investigation:** Jennifer Ramirez, Ibrahim G. Saleh, Evan S. K. Yanagawa, Marie Shimogawa, Emily Brackhahn.

**Methodology:** Jennifer Ramirez, E. James Petersson, Elizabeth Rhoades.

**Project administration:** Jennifer Ramirez, E. James Petersson, Elizabeth Rhoades.

**Resources:** E. James Petersson, Elizabeth Rhoades.

**Supervision:** E. James Petersson, Elizabeth Rhoades.

**Writing – original draft:** Jennifer Ramirez.

**Writing – review & editing:** Jennifer Ramirez, Ibrahim G. Saleh, E. James Petersson, Elizabeth Rhoades.

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
