## [Decision Letter · Decision Letter 0]

26 Nov 2023

PONE-D-23-34232Multivalency drives interactions of alpha-synuclein fibrils with tauPLOS ONE

Dear Dr. Rhoades,

Thank you for submitting your manuscript to PLOS ONE. After careful consideration, we feel that it has merit but does not fully meet PLOS ONE’s publication criteria as it currently stands. Therefore, we invite you to submit a revised version of the manuscript that addresses the points raised during the review process.

We look forward to receiving your revised manuscript.

Kind regards,

Patrick van der Wel, PhD

Academic Editor

PLOS ONE

Journal Requirements:

**Additional Editor Comments:**

The manuscript in its current form fails to comply with multiple of the PLOS ONE publication criteria (https://journals.plos.org/plosone/s/criteria-for-publication). In particular, it fails to meet criterion 3, based on incomplete information on the experimental conditions, a lack of sufficient experimental conditions (and replicates) and concerns about the experimental design more generally. These concerns affect multiple parts of the experiments in the manuscript (see reviewer comments). In addition, there are concerns that the provided data do not fully support the conclusions (and title) of the manuscript (i.e. criterion 4). Although PLOS ONE does not require novelty, it is important that the manuscript properly addresses and discusses previous related publications (criterion 2). You can find a further delineation of concerns, and a request for additional data to be provided, in the reviewer comments. I generally agree with the concerns raised by the reviewers, and we will require a thorough revision of the manuscript (and the data contained therein) and a detailed response to each point raised by the reviewers.  Aside from the comments from the reviewers, please also address the following concerns I have:- Figure 4: I assume different lanes in the SDS PAGE gel are meant to be replicates but they seem to differ a lot. This should be explained/discussed. I would actually advise that this should be repeated, unless I am misunderstanding the nature of the lanes in this gel. (in which case this should be better explained in the caption)- Figure 5: currently the manuscript just notes that 'fibrillar aggregates' are still formed, but does not discuss morphological changes that are apparent in the EM. This needs to be rectified, as morphological changes are indicative of changes in the aggregation mechanism induced by aSYN seeds.- Figure 7 seems to show (unexpected?) inhibitory effects of the aSYN aggregates at the highest concentration. This needs to be discussed in the manuscript. As noted by the reviewers, the current data do not fully support the conclusions, due to limitations in the data and results. Unless these concerns are fully and carefully addressed, I am afraid that the manuscript would not be suitable for publication in PLOS ONE. I therefore stress the importance of a thorough and major revision. Particular attention needs to be paid to the characterization of the seeds, a more detailed analysis of time-dependent aggregation kinetics, a discussion of (and perhaps increased use of) replicates in the various experiments, statistical analysis, FCS analysis, and better explanation of the 'multivalency' evidence from the current study.

Reviewers' comments:

Reviewer's Responses to Questions

**Comments to the Author**

1. Is the manuscript technically sound, and do the data support the conclusions?

Reviewer #1: Partly

Reviewer #2: Partly

Reviewer #3: Partly

2. Has the statistical analysis been performed appropriately and rigorously? 

Reviewer #1: Yes

Reviewer #2: No

Reviewer #3: Yes

3. Have the authors made all data underlying the findings in their manuscript fully available?

Reviewer #1: Yes

Reviewer #2: Yes

Reviewer #3: Yes

4. Is the manuscript presented in an intelligible fashion and written in standard English?

Reviewer #1: Yes

Reviewer #2: Yes

Reviewer #3: Yes

5. Review Comments to the Author

Reviewer #1: The paper studies the interactions between full-length tau and individual tau domains with monomeric and aggregated α-synuclein (αS) using FCS, ThioflavinT, TEM, and charge analysis. The findings suggest a nuanced interplay requiring multivalency, challenging the simplistic notion of direct interaction between tau and alpha-synuclein. The paper addresses a topic of high importance for understanding the origins of mixed pathologies in neurodegenerative disorders. The paper contributes valuable insights into tau-αS interactions, but the authors should address the concerns below to make it suitable for publication.

The conclusion that αS does not affect tau aggregation is questionable due to the relatively coarse time points of ThT fluorescence measurements (every 8-10 hours), potentially missing crucial aggregation windows. The authors should repeat data collection with finer time intervals (10-15 mins) to strengthen the conclusions.

The authors should include paragraph(s) in FCS and ThT section of the Methods detailing the number of technical and biological replicates used for their studies.

The FCS findings are likely dampened due to the heterogeneity of the solutions authors used as seeds. However, generating heterogeneous samples containing a discrete size of oligomers is quite challenging. The authors need to include a paragraph in the Discussion addressing the effect sample heterogeneity on the interpretation of FCS data.

Minor suggestions

1) Add a paragraph in the introduction discussing current in vivo models of synuclein and tau

2) In Fig. 1, provide diagrams of all constructs (tauPRR, tau4R, and tau1N4R) used in the study along with the location of fluorescent tag (tau)

3) In Fig. 1 for synuclein, display the different constructs and label the acetylation and phosphorylation sites.

3) Fig 2, 3 – in the figure captions, add the concentrations of tau constructs and the molar ratios of tau: syn

4) Similar to synuclein, PTMs are implicated in tau aggregation. Add a paragraph explaining the rationale for not using tau with PTM in the results.

5) The quality of TEM micrographs should be revisited – it was difficult to assess whether the low quality is due to the file conversion or the actual data

6) The findings of Figs S3 are important and should be moved to the main Fig 2

7) Figure S7 should be added as a panel in Fig. 5.

Reviewer #2: The manuscript entitled „Multivalency drives interactions of alpha-synuclein fibrils with tau” presents an interesting set of experimental results that could contribute to better understanding of development, prevention and future therapies of neurodegenerative diseases, such as Alzheimer’ disease and Parkinson’s disease. The text is well and clearly written. The conclusions of the work are based mainly on the fluorescence correlation spectroscopy (FCS). The method is related to the hydrodynamic properties of fluorescently labeled species that diffuse freely through a small focal volume. The methodological approaches are described in much detail. The laboratory work is well done, with almost all necessary strict controls (some details are discussed below).

Having said that, however, I would have some serious concerns that would need to be clarified in a revised version of the manuscript.

Major concerns:

1. The motivation to undertake studies of the N-terminal alpha-S 100 residues is unclear. The Authors inform us in the Introduction section that the C-terminal part is important for tau binding, while the N-terminal residues are involved in mediation of binding to lipid bilayers. A scientific question needs to be clearly addressed regarding the putative interaction of the N-terminal fragment of alpha-S with tau in the context of membrane binding.

2. The final conclusions seem to confirm the title "multivalence" only indirectly. In my opinion, the title seems to be a bit too far-reaching compared to the results described in this paper. The results and conclusions should address the multivalency more unambiguously.

3. The FCS approach links the hydrodynamics with spectroscopy, and hence to analyze the hydrodynamic properties in the exact, quantitative manner, it is necessary to take into account the photophysical properties of the fluorescent probe (Alexa Fluor triplet state in this case) together with the diffusive properties of the studied species. Alternatively, the Authors should prove directly that neglecting of the Alexa Fluor triplet state (with the characteristic life-time in the range of microseconds) does not influence the diffusion time determination. Moreover, it is recommended to fit a two-component equation to FCS autocorrelation curves even for a single protein species to check for the purification level from the residual dye present in the solution after labeling, that could significantly change the apparent results.

4. How did the Authors determine, control or check the dimensions or the monomeric/oligomeric/fibril composition of the heterogeneous seed solutions after sonication and the repeatability of the process?

5. The TEM images are below publication quality and should be moved to the Supplement, in particular since they do not contribute significantly to the conclusions.

6. My main concern is, however, related to the statistical data analysis and their interpretation. While the use of the relative diffusion time values derived from FCS is a very good idea, this should be performed with care. First of all, this is only a semi-quantitative “fingerprint” of the aggregation process. The autocorrelation FCS curve, as defined in this work, is related to such a number of diffusing species at equilibrium as defined by the n-component equation. These conditions are not met in the case of aggregation, since (1) the resulting aggregates are heterogeneous (of many different possible types) and (2) they appear during the diffusion time through the focal volume (non-equilibrium state). In particular, the oscillating “serrated pattern” at the longer lag times in S2 Fig. shows this clearly. In consequence, the relative diffusion time values (and the remaining fitting parameters) should be denoted to as “apparent”, since they cannot be ascribed to any given well defined protein complex or state. Moreover, the next question is whether the quantitative statistical parameters such as: median, mean and standard deviation are even applicable to these data, since each data point comes from its own autocorrelation curve with its own uncertainty, which is sometimes tiny, and sometimes huge. Thus, such points cannot be treated as equivalent input for averaging. In my opinion, such a seemingly strict, quantitative statistical analysis is illegitimate and FCS data should be treated only semi-quantitatively as an illustration of the analyzed process.

Minor concerns:

1. Some fragments of the Methods section are repeated, in particular most of the protein expression and purification protocols are common for all constructs. The Authors may consider shortening of the text. Moreover, some buffer solutions of the same composition are denoted to as different names, eg. TBB and ASB3. The nomenclature should be made consistent throughout the text.

2. Was 10% glycerol used for freezing the protein samples?

3. In p. 7, top, the extinction coefficients should be written in a more eligible way: Eps 280 nm = 7450 and 1490 M-1cm-1 for tau 1N4R and tau4R/PRR, respectively.

4. There are some typos, such as “alhough” in p. 5, and repeats “in our”, “set of curves”. The text should be polished.

5. The MALDI mass spectrum is unreadable. The figure should display clearly the deconvoluted spectrum, with the readable axes legends and numbering.

6. The UV spectrum should be equipped with its own legend. This is not a part of the MALDI spectrum. The elevated values of absorbance in the range above 310 nm suggest the presence of the protein aggregates. This should be explained.

Reviewer #3: Comments

The authors investigate interaction between tau and alpha-synuclein (αSyn) proteins associated with neurodegenerative diseases. Using fluorescence correlation spectroscopy, the authors found that tau interacts weakly with monomeric αSyn, but that this interaction is much more pronounced for αSyn aggregates. They also found that the interaction between tau and αSyn does not significantly impact tau aggregation or fibril formation. The authors suggest that the weak and transient interactions between tau monomers and αSyn seeds may act to buffer the monomer tau concentration rather than provide a surface to template tau aggregation.

Although the use of fluorescence correlation spectroscopy techniques to study tau and αSyn interactions is interesting, the findings provide little new information: it is known that both tau and αSyn independently and together form condensates that transition into aggregates.

Major concerns are:

The Materials and Method section on page 9 describes seeds as “seeds’ are expected to be a mix of monomer, oligomer and fiber”, but no characterization has been performed to confirm this assumption.

There are numerous evidence supporting the concept that αSyn aggregates or seeds can promote tau aggregation in vitro, cell culture studies and in vivo (J. Biol. Chem. (2020) 295(21) 7470 –7480; Biological Psychiatry 2018, 84, 499-508; Cell. 2013,154(1):103-17. Sci Rep 12, 2987 (2022); J Exp Med (2021) 218 (1): e20192193.). However, this study did not see acceleration of tau aggregation in the presence of αS seeds. This study relies solely on ThT assay for the aggregation experiments. The SDS-PAGE assay was performed using a single set of conditions in which 1 μM of αSyn seed was mixed with tau1N4R at a 1:5 molar ratio of αSyn to tau1N4R to study the aggregation assay. However, it is essential to conduct further studies on the aggregation using different sets of conditions. In addition to ThT and SDS-PAGE, other assays such as native gel, etc., need to be included to analyze aggregation studies.

In summary, the data seem incomplete, provide not enough novelty, and do not support the conclusion.

6. PLOS authors have the option to publish the peer review history of their article (what does this mean?). If published, this will include your full peer review and any attached files.

Reviewer #1: No

Reviewer #2: **Yes: **Anna Niedzwiecka

Reviewer #3: No

---

## [Author Response · Author response to Decision Letter 0]

10 May 2024

We have uploaded a rebuttal letter as a separate file labeled 'Response to Reviewers' that responds to each point raised by the academic editor and reviewer(s). We have also uploaded a marked-up copy of our manuscript that highlights changes made to the original version, as a separate file labeled 'Revised Manuscript with Track Changes'. We have also uploaded an unmarked version of our revised paper without tracked changes, as a separate file labeled 'Manuscript'.

---

## [Decision Letter · Decision Letter 1]

4 Jul 2024

PONE-D-23-34232R1Multivalency drives interactions of alpha-synuclein fibrils with tauPLOS ONE

Dear Dr. Rhoades,

Thank you for submitting your manuscript to PLOS ONE. After careful consideration, we feel that it has merit but does not fully meet PLOS ONE’s publication criteria as it currently stands. Therefore, we invite you to submit a revised version of the manuscript that addresses the points raised during the review process.

 I apologize for the long delay with this paper, due to a combination of issues. You will see that I had trouble reaching all three original reviewers. One of the original reviewers has provided comments on the current manuscript, noting a number of smaller issues that you may choose to correct. Here I note that PLOS ONE does not provide a pre-publication proof so this is the final chance to make these corrections to your paper (e.g the issues with text size in Fig. S1)

We look forward to receiving your revised manuscript.

Kind regards,

Patrick van der Wel, PhD

Academic Editor

PLOS ONE

Journal Requirements:

Reviewers' comments:

Reviewer's Responses to Questions

**Comments to the Author**

1. If the authors have adequately addressed your comments raised in a previous round of review and you feel that this manuscript is now acceptable for publication, you may indicate that here to bypass the “Comments to the Author” section, enter your conflict of interest statement in the “Confidential to Editor” section, and submit your "Accept" recommendation.

Reviewer #2: (No Response)

2. Is the manuscript technically sound, and do the data support the conclusions?

Reviewer #2: Yes

3. Has the statistical analysis been performed appropriately and rigorously? 

Reviewer #2: No

4. Have the authors made all data underlying the findings in their manuscript fully available?

Reviewer #2: Yes

5. Is the manuscript presented in an intelligible fashion and written in standard English?

Reviewer #2: Yes

6. Review Comments to the Author

Reviewer #2: In the revised version of the manuscript, the Authors have generally addressed most of my concerns, in particular (1) the scientific question has been better defined in the Introduction; (2) the importance of the multivalence interactions has been discussed more profoundly in the last two paragraphs of the Discussion section; (3) the TEM images are now of very good quality, with visible fibril details. Other minor corrections have also been included. Two issues, however, would require some further elaboration.

1. As part of the answer to the question of whether the Alexa 488 triplet state can be neglected, the Authors cite their previous publications. I greatly appreciate the Authors' 20 years of experience in the application of FCS. Unfortunately, none of the articles cited in the response directly address this issue: (1) in Biochemistry 2009, the FCS method was not used at all; (2) in JACS 2012, the FCS results were mentioned, but no autocorrelation curves were shown there, so it is difficult to infer the photophysics of Alexa 488 under those conditions; (3) in JBC, the FCS analysis was performed for objects with diffusion times on the order of 1 ms, so possible effects related to the photophysics of the dye in the microsecond range could be neglected. The Authors also prepared an additional Fig. S3. The statistical model comparison shown therein was carried out in the lagtime range above 10 microseconds, i.e. in the range where this effect is no longer visible by its nature; the result showing that the triplet state does not contribute to the autocorrelation curve in the range above 10 us is thus trivial. The analysis in Fig. S3 is inconsistent with Fig. S4, where the whole data in the range from 1 microsecond are analyzed, and the tauD2app results shown in Fig. 3 seem to vary widely, including very small values. If the Authors believe that the photophysics of Alexa 488 does not affect the final results, this assumption should be written explicitely in the Methods, and the analysis in Fig. S3 and S4 should be presented in a consistent manner.

On the other hand, with regard to the possible presence of a free dye in a solution, the Authors have explained that the single-component diffusion model is sufficient.

2. The mean value and standard deviation are applicable only when they refer to multiple repetitions of the same measurement, a Gaussian random variable, i.e., for example, measurements for monomers. However, in a situation where we are studying aggregation and analyzing the relative change in the diffusion time of the resulting complexes (oligomers, aggregates, higher-order structures), the essence of the experiment is that we do not collect multiple repetitions of measurements for the same object, but allow for the heterogeneity of the sample under study. For a heterogeneous sample, the use of the classical statistics is not adequate, but it is more justified to use so-called robust statistics, in which the robust estimators are the median and quartiles. Since the robust statistics should be applied to the aggregates, all data should be analyzed in the same, consistent way. E.g. box plots would be very useful instead of mean and SD, or more sophisticated violin plots, similar to what the Authors did with great success in their work in JBC 2019.

Moreover, I cannot agree with the Authors’ statement that “some of the reviewer’s concern appears to be misunderstanding. The curves showing the oscillating patterns in S4 Figure are examples of curves that were discarded from our analysis because of our inability to analyze them meaningfully – i.e. the curves that the reviewer points out as problematic were never included in any of the analysis.”, since the curves showing the oscillating patterns are still present in the right-hand column of Fig. S4 (c) and (d) with a legend indicating that these are “the remaining autocorrelation curves following the SSR analysis described in the Materials and Methods”. The diffusion times derived from the autocorrelation curves encompassing the oscillating pattern that reflects the aggregation process occurring during the measurement are determined with much greater uncertainty.

Minor points:

1. In the legend to Fig. S4, a typo should be corrected: there is “SRR”, it should be “SSR”.

2. In the inset of (b) in Fig. S1: the numbers indicating the deconvoluted masses should be larger (now they are illegible).

3. The new figures for the main part of the manuscript now display in darker colors than in the original manuscript, making black lettering on a dark green background (Fig. 1) and points (Fig. 2 and 3) invisible. Also, the gels shown in Fig. 4 are almost black, and thus the results shown on them are invisible. It may be an artifact during the file conversion, but this should be checked.

7. PLOS authors have the option to publish the peer review history of their article (what does this mean?). If published, this will include your full peer review and any attached files.

Reviewer #2: **Yes: **Anna Niedzwiecka

---

## [Author Response · Author response to Decision Letter 1]

29 Jul 2024

Comments are addressed in "Response to Reviewers" document.

---

## [Editor Report · Decision Letter 2]

13 Aug 2024

Multivalency drives interactions of alpha-synuclein fibrils with tau

PONE-D-23-34232R2

Dear Dr. Rhoades,

We’re pleased to inform you that your manuscript has been judged scientifically suitable for publication and will be formally accepted for publication once it meets all outstanding technical requirements.

Kind regards,

Patrick van der Wel, PhD

Academic Editor

PLOS ONE
---

## [Editor Report · Acceptance letter]

2 Sep 2024

PONE-D-23-34232R2 

PLOS ONE

Dear Dr. Rhoades, 

I'm pleased to inform you that your manuscript has been deemed suitable for publication in PLOS ONE. Congratulations! Your manuscript is now being handed over to our production team.

Kind regards, 

on behalf of

Dr. Patrick van der Wel 

Academic Editor

PLOS ONE